# DGAT1 and DGAT2 Inhibitors for Metabolic Dysfunction-Associated Steatotic Liver Disease (MASLD) Management: Benefits for Their Single or Combined Application

**DOI:** 10.3390/ijms25169074

**Published:** 2024-08-21

**Authors:** Miriam Longo, Erika Paolini, Pietro Di Benedetto, Elena Tomassini, Marica Meroni, Paola Dongiovanni

**Affiliations:** Medicine and Metabolic Diseases, Fondazione IRCCS Ca’ Granda Ospedale Maggiore Policlinico, 20122 Milan, Italy; miriam.longo@policlinico.mi.it (M.L.); eryka.paolini.93@gmail.com (E.P.); pietro.dibbe@gmail.com (P.D.B.); ele6895@gmail.com (E.T.); maricameroni11@gmail.com (M.M.)

**Keywords:** MASLD, DGAT, pharmacological therapy, MASH, fibrosis

## Abstract

Inhibiting diacylglycerol acetyltransferase (DGAT1, DGAT2) enzymes (iDGAT1, iDGAT2), involved in triglyceride (TG) synthesis, improves hepatic steatosis in metabolic dysfunction-associated steatotic liver disease (MASLD) patients. However, their potential synergism in disease onset (SLD) and progression (metabolic dysfunction-associated steatohepatitis, fibrosis) has been poorly explored. We investigated iDGAT1 and iDGAT2 efficacy, alone or combined (iDGAT1/2) on fat accumulation and hepatocellular injury in hepatocytes (HepG2) and on fibrogenic processes in hepatic stellate cells (LX2). We further tested whether the addition of MitoQ antioxidant to iDGAT1/2 would enhance their effects. SLD and MASH conditions were reproduced in vitro by supplementing Dulbecco’s Modified Eagle’s Medium (DMEM) with palmitic/oleic acids (PAOA) alone (SLD-medium), or plus Lipopolisaccaride (LPS), fructose, and glucose (MASH-medium). In SLD-medium, iDGAT1 and iDGAT2 individually, and even more in combination, reduced TG synthesis in HepG2 cells. Markers of hepatocellular damage were slightly decreased after single iDGAT exposure. Conversely, iDGAT1/2 counteracted ER/oxidative stress and inflammation and enhanced mitochondrial Tricarboxylic acid cycle (TCA) and respiration. In HepG2 cells under a MASH-like condition, only iDGAT1/2 effectively ameliorated TG content and oxidative and inflammatory mediators, further improving bioenergetic balance. LX2 cells, challenged with SLD/MASH media, showed less proliferation and slower migration rates in response to iDGAT1/2 drugs. MitoQ combined with iDGAT1/2 improved cell viability and dampened free fatty acid release by stimulating β-oxidation. Dual DGAT inhibition combined with antioxidants open new perspectives for MASLD management.

## 1. Introduction

Metabolic dysfunction-associated steatotic liver disease (MASLD) is the primary cause of chronic liver issues in developed nations, affecting roughly 25–30% of the population [1]. MASLD is marked by excessive fat accumulation in the liver, surpassing 5% of organ mass, without alcohol-induced damage and other external factors like viral infections [2].

The prevalence of MASLD has surged due to the rise of obesity and type 2 diabetes mellitus, significantly contributing to conditions like cirrhosis and hepatocellular carcinoma (HCC) [3,4]. MASLD spans from simple fat buildup to metabolic dysfunction-associated steatohepatitis (MASH), potentially leading to fibrosis, cirrhosis, and HCC [5].

Metabolic disorders like glucose intolerance, hypertension, dyslipidemia, and insulin resistance (IR) are intricately linked with MASLD, constituting what is known as metabolic syndrome (MetS) [6]. In addition, several environmental factors, including gender, age, ethnicity, and geography, together with genetic predisposition strongly affect MASLD symptomatology [7].

While lifestyle changes are the cornerstone for MASLD management, at the beginning of 2024, the first pharmacological treatment was approved by the US Food and Drug Administration (FDA) [8]. Resmetirom, a small liver-specific molecule agonist of thyroid hormone receptor β, has proven to be effective in ameliorating the hepatic condition of patients affected by MASH and fibrosis [9]. Indeed, Resmetirom leads to reductions in hepatic fat content and liver stiffness, improvement of non-invasive markers of liver fibrogenesis, and promotion of positive cardiovascular profile by decreasing atherogenic biomarkers [10]. However, while Resmetirom seems to be well tolerated, with only nausea and diarrhea as adverse effects, long-term surveillance is required to exclude potential risks related to thyroid and gonadal axis changes or bone diseases.

Although the approval of the first pharmaceutical treatment for MASLD is a big step forward, Resmetirom efficacy was evident mainly in later MASLD stages, i.e., MASH, due to more severe fibrosis [11]. Therefore, challenges persist in criteria of enrollment, diagnosis, monitoring, and treatment, thus necessitating novel diagnostic tools and therapeutic targets [1]. Since hepatic fat accumulation is the main driver of MASLD progression, enzymes involved in lipogenic pathways might represent viable options to target [12].

Diacylglycerol acetyltransferase 1 and 2 (DGAT1, DGAT2) are enzymes involved in triglyceride (TG) synthesis and operate in the endoplasmic reticulum (ER) [12]. Even if both catalyze the same reaction, condensing diacylglycerol and fatty acyl-CoA to form TGs, they exhibit differences in substrate preference [13]. DGAT1 is mainly involved in free fatty acid (FFA) repackaging in the small intestine and adipose tissue, whereas DGAT2 expression is more liver-specific [13,14,15]. Another difference between the two enzymes is the selectivity for substrates: DGAT1 can transfer fatty acyl-CoA to multiple acceptors in addition to diacylglycerol, including retinol or long-chain alcohols, to form retinyl esters or ceramides, respectively, whereas DGAT2 activity is largely specific to the synthesis of TGs [16,17]. In addition, DGAT2 plays a crucial role in triggering hepatic de novo lipogenesis (DNL) by upregulating SREBP1c [18,19], thus making its inhibition a potential therapeutic target [20]. Indeed, preclinical studies on gene silencing of both DGAT1 and DGAT2 demonstrate promising outcomes. Liver-specific DGAT 1 knock-out as well as its pharmacological inhibition protected against hepatic steatosis and hypertriglyceridemia in mice fed a high-fat diet by reducing TG synthesis and stimulating beta-oxidation [21]. Regarding DGAT2, its hepatic deficiency in a murine MASLD model lowered fat accumulation by reducing TG synthesis and DNL [19]. Accordingly, antisense oligonucleotide against DGAT2 attenuated the expression of genes involved in DNL and lowered TG secretion rate and plasma lipoproteins [20,22,23]. 

The effect of DGAT1 and DGAT2 inhibition has also been evaluated in clinical trials. A randomized, multicenter, double-blind placebo-controlled study in adults with MASLD showed that the DGAT1 inhibitor Pradigastat reduced liver fat content at 24 weeks compared to placebo, although diarrhea affected more than 50% of patients receiving the drug (NCT01811472).

Another adverse effect of DGAT1 inhibitor recorded in clinical trials in metabolic subjects was the increase in FFAs responsible for gastrointestinal inflammation [24,25]. Concerning DGAT2, its inhibition with Ervogastat for 14 days appeared to be well tolerated by healthy adults and by MASLD patients, with robust effects of reducing liver fat and serum triglycerides in a dose-dependent manner, although it did not completely repress but rather dampened their production [26].

Therefore, it would be intriguing to investigate the combination of the two drugs to minimize the side effects of DGAT1 inhibitor (iDGAT1) and potentiate the efficacy of DGAT2 (iDGAT2) in reducing fatty accumulation. To this purpose, in this study we examined the impact of iDGAT1 and iDGAT2 co-treatment (iDGAT1/2) in human hepatoma (HepG2) cells, mimicking SLD/MASH conditions in vitro, and compared their efficacy with the single administration of inhibitors. In addition, we assessed the iDGAT1/2 synergism in counteracting hepatic stellate cell (HSC) activation, which mediate fibrogenesis, by exploiting immortalized human LX2 cells. 

## 2. Results

### 2.1. Determining Concentrations and Efficacy of iDGAT1 and iDGAT2

The first aim of this study was to define the minimum iDGAT1 and/or iDGAT2 concentrations required to reduce hepatocellular fat content. As expected, HepG2 cells treated with 167 µM PAOA accumulated lipid droplets, evidenced by Oil Red O (ORO) staining (Appendix A). iDGAT1 and iDGAT2 treatments alone reduced TG content alongside DGAT1 and DGAT2 mRNA levels without showing significant differences across the tested doses (25, 50, 100 µM) (Appendix A). 

In the attempt to choose the ideal iDGAT1/2 concentration, we performed a dose-response curve testing the combination of the two inhibitors at 25, 50, and 100 µM. At MTS assay, we observed that only iDGAT1/2 at 25 µM improved cell viability (Appendix A; * *p* < 0.05 vs. CT, iDGAT1/2 at both 50 and 100 µM). In keeping with these results, we decided to combine the two inhibitors at the lowest concentration (25 µM) for the following experiments. 

### 2.2. iDGAT1/2 Treatment Reduces TG Synthesis and DNL in HepG2 Cells Incubated with SLD-Medium

DGAT1 and DGAT2 levels were increased in HepG2 cells treated with PAOA (Figure 1A,B; * *p* < 0.05 vs. CT). Accordingly, intracellular TG content was enhanced compared to controls (Figure 1C; ** *p* < 0.01 vs. CT). The combination of iDGAT1/2 exhibited a greater efficacy in inhibiting TG synthesis, by reducing DGAT1/2 expression (Figure 1A,B; * *p* < 0.05 vs. PAOA) and intracellular TG content (Figure 1C; ** *p* < 0.01 vs. PAOA) compared to the single inhibitors (Figure 1A–C; * *p* < 0.05 vs. iDGAT1 and iDGAT2). In addition, iDGAT1/2 alone and even more in combination dampened the expression of SREBP1 and SREBP2, key players in DNL (Figure 1D; * *p* < 0.05 vs. PAOA, iDGAT1 and iDGAT2). These findings support that iDGAT1/2 ameliorate fat accumulation in hepatocytes by downregulating both TG synthesis and DNL in a more efficacious way than the single treatments. 

#### 2.2.1. iDGAT1/2 Administration Attenuates Inflammation and Oxidative Stress in HepG2 Cells under SLD-Medium

We observed that the use of iDGTA1/2 reduced lipid content in HepG2 cells, especially when administrated in combination. Therefore, we assessed their efficacy in counteracting PAOA-mediated hepatocellular injury. As expected, PAOA treatment induced ER and oxidative stress as well as inflammation by stimulating the release of cytokines in HepG2 cells (Appendix A; * *p* < 0.05 vs. CT).

The supplementation of iDGAT1 and iDGAT2 alone or in combination reduced activating transcription factor 4 (ATF4) and X-box binding protein 1 (XBP1) gene expression, markers of ER stress, and malondialdehyde (MDA) levels, a byproduct of lipid peroxidation, with a similar efficacy (Appendix A; ** *p* < 0.01 vs. PAOA). Conversely, only iDGAT1/2 counteracted oxidative stress by decreasing manganese-dependent superoxide dismutase 2 (MnSOD2) mRNA levels and intracellular ROS (Appendix A; * *p* < 0.05 vs. PAOA).

Similarly, iDGAT1/2 administration had a greater effect in lowering the release of pro-inflammatory cytokines (TNFα, IL-1α, IL-6, MCP-1) compared to the single treatments (Appendix A–H; * *p* < 0.05 vs. PAOA). 

These findings suggest that single DGAT inhibitors are efficacious against fat accumulation and lipid peroxidation, but they partially impact on oxidative stress. Conversely, their combination is the best strategy for mitigating PAOA-induced cell damage and inflammatory response, thus making their application suitable for the management for MASH.

#### 2.2.2. Functional Role of iDGAT1 and iDGAT2 in Mitochondrial Activity Restoration in HepG2 under SLD-Medium

Fat overload represents the primary hit for mitochondrial dysfunction and disease progression in MASLD. To date, the impact of DGAT inhibitors on mitochondrial activity has not been described. We observed that the exposure of HepG2 cells to fatty acids reduced citrate synthase (CS) activity (Figure 2A; * *p* < 0.05 vs. CT) and the NAD+/NADH ratio (Figure 2B; ** *p* < 0.01 vs. CT), suggesting an impairment of TCA. Although iDGAT1 and iDGAT2 administered alone showed poor efficacy in improving mitochondrial TCA, the combination of iDGAT1 and -2 completely restored CS activity (Figure 2A; * *p* < 0.05 vs. PAOA) and significantly elevated NAD+/NADH levels (Figure 2B; ** *p* < 0.01 vs. PAOA). No differences were found in the activities of mitochondrial complexes I and III in HepG2 cells incubated with PAOA and iDGAT treatments (Figure 2C,D). Interestingly, ATP synthase activity increased with iDGAT1/2 (Figure 2E; ** *p* < 0.01 vs. PAOA) and was paralleled by a significant increase in the ATP/ADP ratio and total ATP content (Figure 2F,G; * *p* < 0.05 vs. PAOA). Moreover, iDGAT1/2 decreased lactate levels (Figure 2H; * *p* < 0.05 vs. PAOA). In sum, our results support the notion that iDGAT co-treatment recovered aerobic respiration, which is impaired by fatty acid loading. 

### 2.3. iDGAT1/2 Treatment Reduces TG Synthesis and DNL in HepG2 Cells Incubated with MASH Medium

We next examined the effect of iDGAT inhibitors on disease progression by mimicking in vitro the MASH condition. Similar to SLD, HepG2 cells accumulated intracellular lipids, as revealed by ORO staining (Figure 3A) and enhanced TG synthesis as observed by the elevated expression of DGAT enzymes (Figure 3B–D; * *p* < 0.05 vs. CT), along with DNL (Figure 3E; ** *p* < 0.01 vs. CT).

In MASH medium, the individual drugs failed to decrease lipid droplet content, and this effect was possibly due to lack of inhibition of SREBP1/2 (Figure 3E). Conversely, combined iDGAT1/2 treatment strongly diminished fat accumulation in ORO staining (Figure 3A), DGAT1 and DGAT2 mRNA and protein levels (Figure 3B,C; ** *p* < 0.01 vs. MASH), intracellular TG content (Figure 3D; ** *p* < 0.01 vs. MASH), and DNL (Figure 3E; * *p* < 0.05 vs. MASH).

#### 2.3.1. Impact of iDGAT1 and iDGAT2 on Inflammatory Response in HepG2 Cells Treated with MASH-Medium

Emerging evidence has supported the contribution of DGAT inhibitors administered alone in reducing intracellular TG content. However, their potential beneficial effects on ER and oxidative stress, inflammation, and mitochondrial dysfunction, which are hallmarks of disease progression, have been poorly investigated either alone or in combination. Compared to an SLD-like in vitro model, HepG2 cells in MASH medium showed an exacerbated expression of ATF4 and XBP1 genes (Appendix A; ** *p* < 0.01 vs. CT), alongside elevated levels of MDA, MnSOD2, and ROS (Appendix A–D; * *p* < 0.05 vs. CT). Accordingly, the secretion of pro-inflammatory cytokines (TNFα, IL-1β, IL6, MCP1) was boosted by MASH medium (Appendix A–H; * *p* < 0.05 vs. CT), thus suggesting that in this condition, HepG2 cells displayed more severe hepatocellular damage than those treated with SLD medium. 

When administered individually, iDGAT1 partially restored ATF4 mRNA levels (Appendix A; ** *p* < 0.01 vs. MASH), whereas it did not affect XBP1, MnSOD2, and lipotoxic intermediates (Figure 3B–D). Conversely, treatment with iDGAT2 failed to dampen ER and oxidative stress. However, the combined treatment with iDGAT1/2 significantly reduced the expression of ATF4 and XBP1 (Appendix A; ** *p* < 0.01 vs. MASH), and decreased MDA, MnSOD2, and ROS levels (Appendix A–D; ** *p* < 0.01 vs. MASH), thus effectively mitigating oxidative stress. Notably, iDGAT1/2 treatment also strongly counteracted the release of pro-inflammatory mediators (Appendix A–H; ** *p* < 0.01 vs. MASH).

#### 2.3.2. iDGAT1/2 Treatment Restores Mitochondrial Activity in HepG2 Cells Incubated with MASH Medium

Mitochondrial dysfunction plays a key role in the switch from simple steatosis to MASH. To date, no studies have explored the effects of DGAT inhibitors on mitochondrial bioenergetics and activity in a condition that mirrors human MASH. Exposure of HepG2 cells to MASH medium significantly reduced CS activity (Figure 4A; * *p* < 0.05 vs. CT) and the NAD+/NADH ratio compared to untreated cells (Figure 4B; * *p* < 0.05 vs. CT). Single treatments minimally affected CS activity (Figure 4A), whereas iDGAT1 and iDGAT2 increased the NAD+/NADH ratio (Figure 4B; ** *p* < 0.01 vs. MASH). The combination of iDGAT1/2 was the only treatment that enhanced CS activity (Figure 4A; * *p* < 0.05 vs. MASH) and increased the NAD+/NADH ratio, akin to individual iDGAT1 and iDGAT2 supplementation (Figure 4B; ** *p* < 0.01 vs. MASH). 

After MASH medium exposure, HepG2 cells showed higher activity of complexes I and III (Figure 4C,D; ** *p* < 0.01 vs. CT), whereas ATP synthase remained unchanged compared to untreated cells (Figure 4E). The reduction in ATP intracellular content (Figure 4F,G; * *p* < 0.05 vs. CT) was paralleled by a tenfold rise in lactate levels (Figure 4H; ** *p* < 0.01 vs. CT) in HepG2 cells challenged with MASH medium, thus suggesting that the latter worsens mitochondrial dysfunction by impairing the respiratory chain with a consequent ROS accumulation and a switching towards anaerobic metabolism. 

Single treatments did not improve mitochondrial respiration. Conversely, iDGAT1/2 supplementation restored complexes I and III activity (Figure 4C,D; * *p* < 0.05 vs. MASH), enhanced ATP synthesis (Figure 4E–G; * *p* < 0.05 vs. MASH), and decreased intracellular lactate levels (Figure 4H; ** *p* < 0.01 vs. MASH). These data supported that iDGAT1/2 co-treatment was the best option for ameliorating hepatocellular damage and mitochondrial dysfunction in MASH. 

### 2.4. iDGAT1/2 Supplementation Mitigates Fibrotic Pathways and LX2 Cell Activation under SLD and MASH Media

In keeping with the results obtained from HepG2 cells, we further examined the impact of DGAT inhibitors on fibrogenic mechanisms, an aspect completely uncharted. To this purpose, we exposed LX2 cells to SLD and MASH-conditioned media. LX2 cells increased the expression of collagen type I alpha 1 chain (Col1A1; Appendix A; ** *p* < 0.01 vs. CT) and hydroxyproline production (Appendix A; ** *p* < 0.01 vs. CT), major components of the extracellular matrix (ECM), and the effect was exacerbated in MASH medium.

According to the increased ECM deposition, the mRNA levels of metalloproteinases 4 (MMP4) and 9 (MMP9), which regulate matrix degradation, were reduced in both media, with the strongest effect due to the MASH one (Appendix A; * *p* < 0.05 vs. CT). In parallel, TIMP metallopeptidase inhibitors 1 (TIMP1) and 2 (TIMP2), which are MMP inhibitors, increased in the presence of SLD and MASH medium (Appendix A; * *p* < 0.05 vs. CT), thus suggesting that in both conditions, an activation of hepatic stellate cells occurs. iDGAT treatments showed a similar efficacy in reducing Col1A1 and hydroxyproline levels in LX2 cells (Appendix A–D; * *p* < 0.05 vs. SLD or MASH media). Concerning ECM degradation, only the iDGAT1/2 supplementation promoted MMP4/9 expression and significantly reduced TIMP1/2 mRNA levels (Appendix A–H; * *p* < 0.05 vs. SLD or MASH media). 

To further confirm LX2 activation and the effect of DGAT inhibitors under SLD and MASH triggers, we performed immunocytochemistry analysis, invasion, and scratch tests. As expected, LX2 cells stimulated with steatotic and even more with MASH media heightened alpha-smooth muscle actin (α-SMA) expression and enhanced proliferation and migration rate (Figure 5A–E). iDGAT1 and iDGAT2 mitigated α-SMA protein levels and delayed LX2 cell growth and invasion, and the effect was most evident with the combination of the two drugs (Figure 5A–E).

To sum up, the single administration of DGAT inhibitors partially counteracts ECM deposition and modulates hepatic stellate cells activation. Notably, these beneficial effects persist when the pharmacological treatments are administered together, thus supporting for the first time their potential application in advanced MASLD. 

### 2.5. Efficacy of MitoQ as Supplementary Treatment in Clearing FFA Outflow

Possible side effects ascribable to DGAT inhibitors are due to excessive FFA release in the circulation, as they escape beta-oxidation. Thus, we hypothesized that the addition of the antioxidant MitoQ to the iDGAT1/2 cocktail at 25 µM, which resulted in the best approach in both HepG2 and LX2 models, could overcome this limitation from the earliest MASLD stages. In the MTS assay, we observed that MitoQ enhanced HepG2 cell viability when combined with iDGAT1/2 in SLD medium after 24 up to 72 h (Figure 6A; * *p* < 0.05 vs. iDGAT1/2).

After the exposure to SLD medium, iDGAT1/2 did not dampen FFA secretion (Figure 6B), possibly due to the fact that they did not stimulate carnitine palmitoyltransferase 1A (CPT1) or peroxisome proliferator activated receptor alpha (PPARα) (Figure 6C,D), two key genes involved in lipid metabolism and fatty acid oxidation. Conversely, by adding MitoQ to the two drugs, HepG2 cells reduced FFA efflux (Figure 6B; ** *p* < 0.01 vs. iDGAT1/2) and upregulated CPT1 and PPARα genes (Figure 6C,D; * *p* < 0.05 vs. iDGAT1/2). 

## 3. Discussion

Lifestyle changes represent the recommended intervention for the management of MASLD and currently only one pharmacological treatment named Resmetirom has been approved showing efficacy in MASH and severe fibrosis [10,27]. Although Resmetirom treatment was successful, its placebo-subtracted effect was overall modest, thus indicating that approximately two or one of ten patients treated will have MASH resolution and fibrosis improvement, respectively. Therefore, complementary therapies will be required to counteract the metabolic comorbidities and the heterogeneous phenotypes simultaneously occurring in most MASLD patients [11].

Since excess TG in the liver is the primary event leading to MASLD onset and progression, the modulation of lipogenic enzymes effectively reduced hepatic fat content in preclinic studies and clinical trials [12]. Specifically, genetic deletion or pharmacological DGAT1 and DGAT2 inhibition reduced hepatic steatosis and hypertriglyceridemia in murine models and MASLD patients (NCT01811472) [19,20,21,22,23,26]. However, no current studies have specifically evaluated the synergistic inhibition of DGAT enzymes in MASLD and its progressive forms. In this study, we assessed the efficacy of DGAT1 and DGAT2 inhibitors, both individually and in combination, on lipid stores, inflammatory responses, mitochondrial activities, and fibrogenic processes by modeling in vitro the conditions of SLD and MASH. 

We first determined the dosage of DGAT1 and DGAT2 inhibitors which can effectively reduce fat accumulation in HepG2 cells exposed to a fatty acid mixture (SLD-medium). We found that iDGAT1 and iDGAT2 administered alone exhibited a high inhibitory activity on TG handling, and this effect was dose-independent. To further establish the optimal concentration at which the DGAT inhibitors can be co-administered, a dose-response curve testing the combination of the two drugs at 25–50–100 µM was performed. Our data revealed that iDGAT1/2 at the lowest dose improved cell viability, thus allowing us to pursue the aims of the study using the minimum dose of inhibitors either alone or in combination. In the SLD model, single DGAT inhibitors moderately reduced lipid content and mitigated lipotoxicity in HepG2, thus confirming the results obtained in previous studies [19,20,21,22,23,26]. However, these effects were lost in hepatocytes exposed to a MASH-like condition.

In addition, their combined supplementation was most effective in lowering lipid content, not only by directly targeting TG synthesis but also by downregulating DNL. Accordingly, iDGAT1/2 supplementation restored markers of oxidative and ER stress as well as pro-inflammatory mediators both in a milder condition (SLD) and even more so during a stronger stimulation (MASH), thereby supporting that dual-enzymatic suppression rather than the single one is the best strategy to attenuate hepatocellular damage. In a placebo-controlled randomized phase 2a trials, Calle et al. revealed that the co-administration of acetyl-coenzyme A carboxylase (ACC) and DGAT2 inhibitors improved liver enzymes and MASH biomarkers, thus restricting the adverse effect observed in patients receiving iACC monotherapy [28].

Unexpectedly, an innovative result emerged from this study regarding the impact of iDGAT1/2 co-treatment on mitochondrial bioenergetic balance. Indeed, the dual administration enhanced mitochondrial respiration and ATP synthesis, thus ensuring the outflow of lipid substrate towards energy production.

We then proceeded to explore the potential efficacy of DGAT1 and DGAT2 inhibitors on fibrosis mechanisms in human LX2 stellate cells which activated and produced ECM in response to SLD and MASH conditioned media. Our data indicated that DGAT inhibitors halted LX2 migration and proliferative rates and reduced ECM synthesis, showing the largest effect with the drug cocktail. In keeping with this findings, Yamaguchi and collaborators previously described that DGAT1 antisense oligonucleotide dampened HSC activation in methionine and choline deficient (MCD)-fed rats, retaining vitamin A and collagen expression [23]. Similarly, iACC orally administered in choline-deficient, amino acid-defined, high-fat diet (CDAHFD)-fed rats reduced liver stiffness and α-SMA protein levels [28]. Conversely, Yenilmez et al. revealed that ob/ob mice receiving RNA interference targeting DGAT2 ameliorated liver fat content without improving histological inflammation and fibrosis [29]. Of note, iDGAT2 combined with iACC attenuated markers of HSC activation and fibrotic scars more than monotherapies [28]. These observations may be explained by the fact that both DGAT1 and ACC are markedly expressed in HSCs, while hepatocytes express the DGAT2 enzyme more [23,29], thus corroborating the hypothesis that multi-drug approaches may be effective against MASH and fibrosis. 

In the last part of the study, we focused on alternative strategies to address the challenges that emerged with the use of DGAT inhibitors, such as excessive FFA release (NCT01811472) [26]. As mentioned above, the use of iDGAT1/2 at 25 µM compared to 50 and 100 µM improved cell viability, which allowed us to administer the lowest concentration in all in vitro models. Although this dose reduction might mitigate adverse effects, our measurements revealed that FFA levels in the cell supernatants did not decrease in HepG2 cells treated with iDGAT1/2, likely due to the inhibitors’ failure to stimulate fatty acid degradation. Thus, we postulated that adding an agent targeting mitochondrial functions (MitoQ) to the iDGAT1/2 cocktail could enhance treatment efficacy. This decision was supported by previous observations which highlighted that antioxidant compounds can provide benefits for MASH treatment and metabolic syndrome [30,31,32]. Here, we demonstrated that MitoQ supplemented with iDGAT1/2 enhanced cell viability and beta-oxidation with a parallel reduction in FFA secretion, showing a greater efficacy compared to the dual regimen. 

In sum, our study demonstrates that the combination of DGAT1 and DGAT2 inhibitors effectively mitigates fat accumulation and hepatocellular damage in hepatocytes, and hampers fibrogenic pathways in HSC cells under both SLD- and MASH-mimicked conditions. The addition of MitoQ further improves these outcomes, suggesting that a triple combination therapy could offer significant benefits for treating MASLD advanced forms. However, we acknowledge that the study has the limitation of having been exclusively carried out in vitro, thus requiring further confirmation in preclinical and human studies. Moreover, the long-term safety and potential side effects of these treatments in combination remain to be examined. Otherwise, the novelty of this work lies in the discovery of new insights regarding the effects of iDGAT on mitochondrial bioenergetic balance in hepatocytes and provides an in-depth characterization of their efficacy on fibrotic processes. We also propose for the first time alternative strategies to bypass the restrictions related to the use of iDGAT inhibitors, stressing the potential application of multi-targeted therapies for MASLD management and paving the way for future clinical investigations. 

## 4. Materials and Methods

### 4.1. Cell Models and Treatments

HepG2 hepatocellular carcinoma cells, resembling primary hepatocyte metabolism, and LX2 human hepatic stellate cells were cultured in DMEM with 10% FBS, 1% L-Glut, and 1% P/S in a 5% CO_2_ humidified incubator at 37 °C. A dose response curve was performed on HepG2 exposed to palmitic and oleic acids (PAOA) 167 µM (1:2 ratio) for 24 h to induce fat accumulation. Then, we tested 25, 50, and 100 µM of iDGAT1 (PF-04620110; Sigma-Aldrich, St. Louis, MO, USA) and iDGAT2 (PF-06424439; Sigma-Aldrich, St. Louis, MO, USA) alone or in combination to define the minimum concentration required to reduce lipid content.

To assess the iDGAT1/2 synergism in dampening TG accumulation and fibrogenic processes, we reproduced in vitro SLD/MASH conditions in both cell lines. HepG2 were supplemented with DMEM enriched in 167 µM PAOA for 24 h to mimic SLD (SLD medium). For MASH, HepG2 cells were treated for 5 consecutive days with DMEM containing 167 µM PAOA, 22.5 mM glucose and fructose, and 5 µM LPS (MASH medium). We collected SLD and MASH media from HepG2 cell cultures and exposed LX2 cells to evaluate the effect of the two inhibitors on fibrotic mechanisms. 

Attempting to enhance the efficacy of DGAT inhibitors, the antioxidant Mitoquinone mesylate (MitoQ), a synthetic analogue of coenzyme Q10, was added to iDGAT1/2 (25 µM) cocktail at a final concentration of 1 µM in HepG2 cells challenged with SLD medium. 

### 4.2. Oil Red O (ORO) Staining

Cells (5 × 10^5^/well) were plated in duplicate on 6-well plates and incubated overnight in DMEM (10% FBS, 1% L-glutamine, 1% penicillin/streptomycin). After 24 h, growth medium was replaced with quiescent medium (0.5% BSA, 1% L-glutamine, 1% penicillin/streptomycin) for another 24 h. Subsequently, ORO staining was performed. Cells were fixed with 4% formalin, washed, and treated with 60% isopropanol. ORO working solution (1 mL/well) was added for 40 min. After rinsing, lipid droplets were visualized in red in brightfield microscopy, and the ORO-positive area was acquired at 20× magnification.

### 4.3. Gene Expression Analysis

RNA was extracted from cell cultures using TRIzol reagent (Life Technologies–ThermoFisher, Carlsbad, CA, USA). Total RNA (1 μg) was retrotranscribed with the VILO random hexamers synthesis system (Life Technologies–ThermoFisher, Carlsbad, CA, USA). Quantitative real-time PCR was performed by an ABI 7500 fast thermocycler using SYBR Green chemistry (Fast SYBR Green Master Mix; Life Technologies–ThermoFisher, Carlsbad, CA, USA). All reactions were delivered in quadruplicate. Data were normalized to the β-actin housekeeping gene, and results are expressed as mean and standard deviation (SD) and graphed as fold increase (arbitrary units, AU). Primers are listed in Table 1. 

### 4.4. Western Blot Analysis

Total protein lysates were extracted from cell cultures, using RIPA buffer containing 1 mmol/L Na-orthovanadate, 200 mmol/L phenylmethyl sulfonyl fluoride, and 0.02 μg/μL aprotinin. Samples were pooled prior to electrophoretic separation, and all reactions were performed in duplicate. Then, equal amounts of proteins (50 μg) were separated by SDS- PAGE, transferred electrophoretically to nitrocellulose membrane (BioRad, Hercules, CA, USA), and incubated with specific antibodies overnight. At least three independent lots of freshly extracted proteins were used for experiments. The antibodies and dilutions used are listed in Table 2.

### 4.5. Immunocytochemistry (ICC)

A total of 1× 10^5^ cells were seeded on coverslips lodged in a 6-well plate in duplicate and kept overnight in DMEM containing 10% FBS, 1% L-glutamine, and 1% penicillin/streptomycin. After 24 h, growth medium was removed and cells were kept for 24 h in quiescent medium, containing 0.5% bovine serum albumin (BSA), 1% L-glutamine, and 1% penicillin/streptomycin.

Next, LX2 cells were fixed in 4% formalin for 10 min and permeabilized in 0.3% Triton X-100 (Sigma-Aldrich, St. Louis, MO, USA). Cells were incubated in 5% BSA for 30 min and with anti-αSMA antibody (Table 2) overnight at 4 °C. Then, each sample was incubated with anti-rabbit horseradish-peroxidase-conjugated antibody, and 3,3′-diaminobenzidine was provided as chromogen. Nuclei were counterstained with hematoxylin. Finally, samples were mounted with a drop of aqueous VectaMount AQ Mounting Medium (Maravai LifeSciences, Inc, San Diego, CA, USA). 

### 4.6. ELISA Assays in Live Cells 

The ATP/ADP ratio and intracellular ROS production were measured through the ADP/ATP assay (MAK135 and MAK144, respectively; Sigma Aldrich, St. Louis, MO, USA) in live cells to evaluate cellular metabolism, mitochondrial activity, and oxidative stress in response to treatments. HepG2 cells were plated (4 × 10^4^ cells/well) and incubated with 90 µL of growth medium in a 96-well plate. The following day, they were treated with SLD or MASH media with or without the addition of the DGAT inhibitors.

### 4.7. ELISA Assays in Cell Lysates 

HepG2 and LX2 cells (106 cells/condition) were incubated with 5% Nonidet (NP-40) at 99 °C, a dense detergent capable of lysing cell membranes. Afterwards, the samples were cooled and centrifuged at maximum speed for 10 min at 4 °C. The supernatants were then transferred to a new tube. NP-40 cell lysates were used with the following kits:-Triglyceride Quantification Assay (MAK266, Sigma Aldrich, St. Louis, MO, USA)-Lipid Peroxidation (MDA) Colorimetric Assay (ab118970, Abcam, Cambridge, UK)-Citrate Synthase Activity Assay (MAK193, Sigma Aldrich, St. Louis, MO, USA)-Mitochondrial Complex I Activity Colorimetric Assay (ab287847, Abcam, Cambridge, UK)-Mitochondrial Complex III Activity Assay (ab287844, Abcam, Cambridge, UK)-ATP Synthase Enzyme Activity Assay Kit (ab109714, Abcam, Cambridge, UK)

Cell supernatants from HepG2 cells were collected for the following measurements:-Il1β/IL-F2 ELISA assays (DLB50, Biotechne R&D Systems, Minneapolis, MN, USA)-IL6 ELISA assays (DLB50, Biotechne R&D Systems, Minneapolis, USA)-TNF-α ELISA assays (DLB50, Biotechne R&D Systems, Minneapolis, USA)-MCP1 ELISA assays (DLB50, Biotechne R&D Systems, Minneapolis, USA)

Cell supernatants from LX2 cells were collected for quantifying extracellular hydroxyproline levels through MAK357 (Sigma Aldrich, St. Louis, MO, USA).

### 4.8. Cell Viability and Proliferation Assay

Cells (2.5 × 10^2^/well) were seeded in a 96-well plate in quadruplicate and incubated in DMEM containing 10% FBS, 1% L-glutamine, and 1% penicillin/streptomycin overnight. Cell proliferation was measured at baseline using the CellTiter96-Aqueous One Solution Cell Proliferation Assay (MTS:3-(4,5-dimethylthiazol-2-yl)-5-(3-carboxymethoxyphenyl)-2-(4-sulfophenyl)-2H-tetrazolium)) kit (Promega Corporation, Fitchburg, WI, USA). Fresh growth medium was provided for 24–48–72 h. MTS reagent (20 μL/well) was added to the cells, followed by incubation for 4 h in a 5% CO_2_ humidified incubator at 37 °C, and the absorbance was measured at 490 nm, at 0, 24, 48, and 74 h. At least 3 independent experiments were performed.

### 4.9. Invasion Assay

LX2 cells were plated on the apical side of a Transwell (12 mm in diameter and 0.8 μm porosity; Transwell, Corning Inc., Lowell, MA, USA) on a collagen-coated polycarbonate filter with a density of 2.5 × 10^4^ cells/well. They were kept growing for 24 h with DMEM supplemented with 10% FBS, 1% L-glut, and 1% P/S on both sides, apical and basolateral. On the second day, quiescent medium (DMEM, 1% L-glut, 1% P/S, and 0.5% BSA) was added in the apical compartment, while SLD or MASH conditioned media with or without DGAT inhibitors was added to the basolateral side to induce migration and to assess the efficacy of the drugs. After 48 h of treatment, the filter was removed, and nuclei were labeled with hematoxylin (Sigma-Aldrich, St. Louis, MO, USA). Finally, the samples were mounted with a drop of aqueous VectaMount AQ Mounting Medium (Maravai LifeSciences, Inc., San Diego, CA, USA). 

### 4.10. Scratch Assay

LX2 cells were plated on a 6-well plate (6 × 10^5^ cells/well) and incubated with DMEM supplemented with 10% FBS, 1% L-glut, and 1% P/S. The following day, LX2 cells were treated with SLD or MASH conditioned media with or without DGAT inhibitors. To measure the migratory capacity of LX2, a scratch was made using a sterile 10 µL pipette tip in the cell monolayer. The opening was photographed with a light microscope (10× magnification) at time 0 and after 24 and 48 h.

### 4.11. Statistical Analysis

Statistical analyses were conducted with Prism (version 6, GraphPad Software, Boston, MA, USA), using the one-way nonparametric ANOVA (Kruskal-Wallis followed by post hoc 2-tailed *t*-test) when two groups were compared, or the Dunn multiple comparison test when multiple groups were compared, adjusted for the number of comparisons. Values of *p* < 0.01 and *p* < 0.05 (2-tailed) were considered statistically significant. 

## Figures and Tables

**Figure 1 ijms-25-09074-f001:**
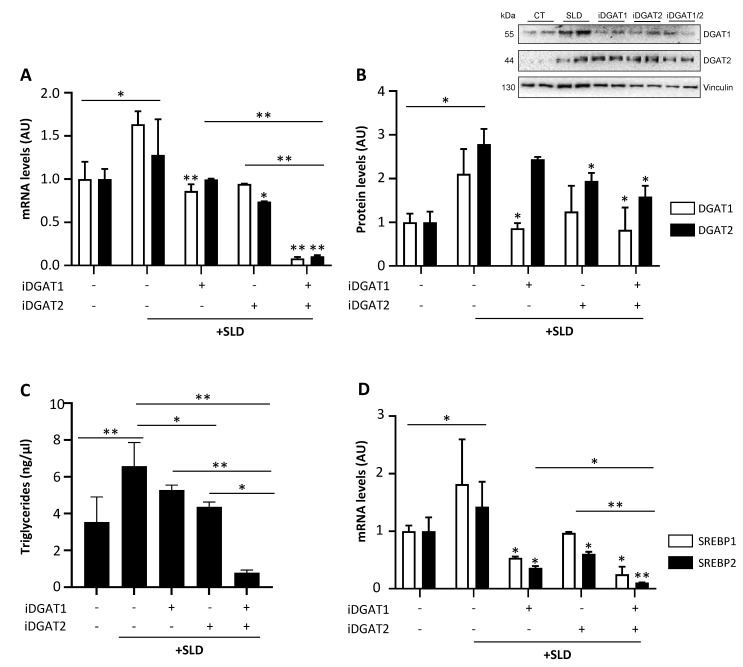
**iDGAT1/2 downregulates TG synthesis and DNL in HepG2 cells under a steatotic liver disease (SLD)-like condition.** DGAT1 and DGAT2 gene expression (**A**) analysis and protein levels (**B**) were assessed in HepG2 cells after challenging with SLD medium with or without iDGAT1, iDGAT2, or both at 25 µM. Triglyceride content (**C**) was measured in HepG2 cell lysates through a colorimetric assay. SREBP1 and SREBP2 mRNA levels (**D**) were assessed by qRT-PCR in HepG2 cells treated with SLD medium and/or DGAT inhibitors alone or in combination. Each experiment was carried out in triplicate. Adjusted * *p* < 0.05 or ** *p* < 0.01.

**Figure 2 ijms-25-09074-f002:**
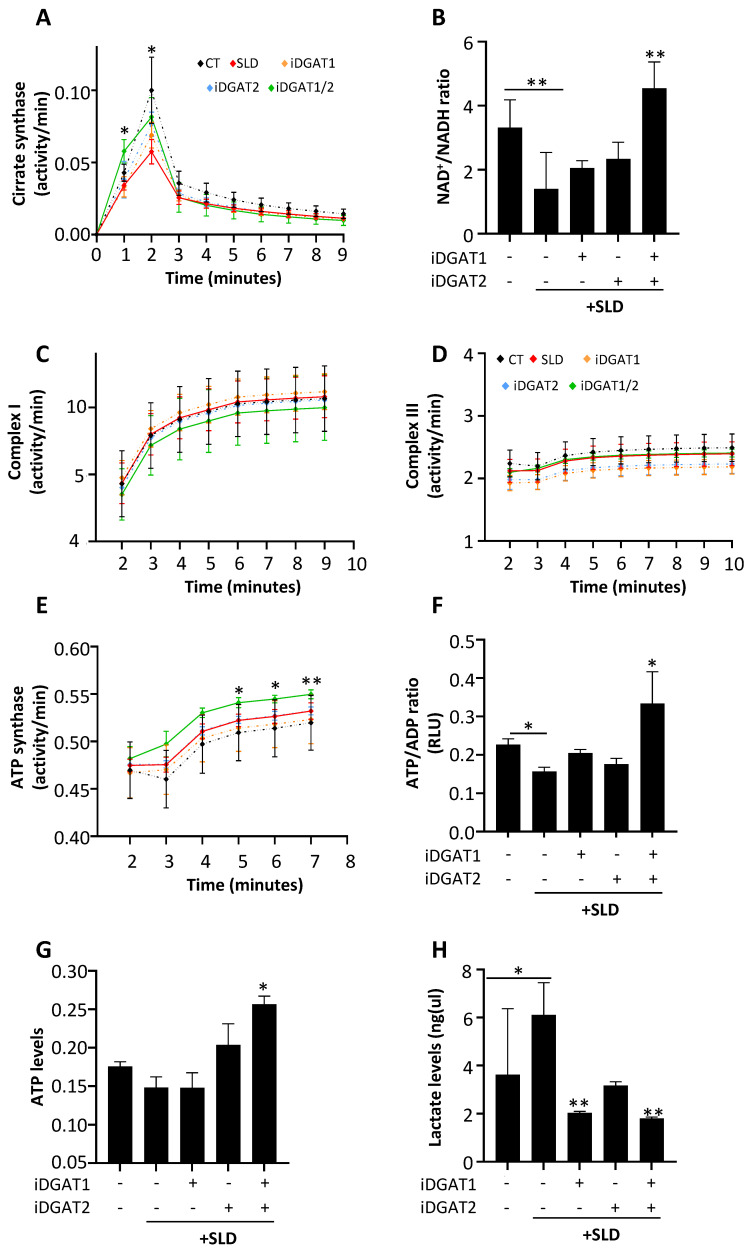
iDGAT1/2 rescued mitochondrial bioenergetics and promoted aerobic respiration in HepG2 cells under an SLD-like condition. Evaluation of TCA was performed by assessing citrate synthase (CS) activity (**A**) over time and measuring the NAD+/NADH ratio (**B**) in HepG2 cells challenged with SLD medium with or without DGAT inhibitors alone or in combination. The mitochondrial respiration rate of HepG2 cells was evaluated through the assessment of complex I (**C**), complex III (**D**), and ATP synthase (**E**) activities. ATP/ADP production was estimated in live HepG2 cells (**F**), while total ATP (**G**) and lactate (**H**) contents were quantified in HepG2 cell lysates after the exposure to SLD medium and/or DGAT inhibitors. Each experiment was carried out in triplicate. Adjusted * *p* < 0.05 or ** *p* < 0.01.

**Figure 3 ijms-25-09074-f003:**
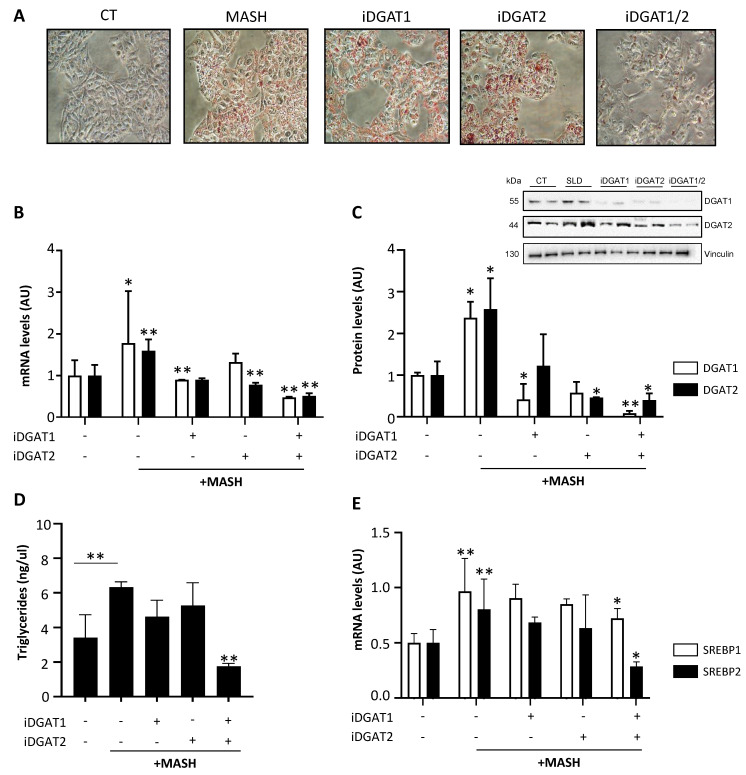
**iDGAT1/2 downregulated TG synthesis and DNL in HepG2 cells under a MASH-like condition.** Representative images (20× magnification) obtained after ORO staining (**A**) showing lipid droplets accumulation in HepG2 cells after MASH medium exposure and the effects of DGAT inhibitors in reducing lipid content. DGAT1 and DGAT2 gene expression (**B**) analysis and protein levels (**C**) were assessed in HepG2 cells after challenging with MASH medium for 5 consecutive days with or without iDGAT1, iDGAT2, or both at 25 µM. Triglyceride content (**D**) was measured in HepG2 cell lysates through a colorimetric assay. SREBP1 and SREBP2 mRNA levels (**E**) were assessed through qRT-PCR in HepG2 cells treated with MASH medium and/or DGAT inhibitors alone or in combination. Each experiment was carried out in triplicate. Adjusted * *p* < 0.05 or ** *p* < 0.01.

**Figure 4 ijms-25-09074-f004:**
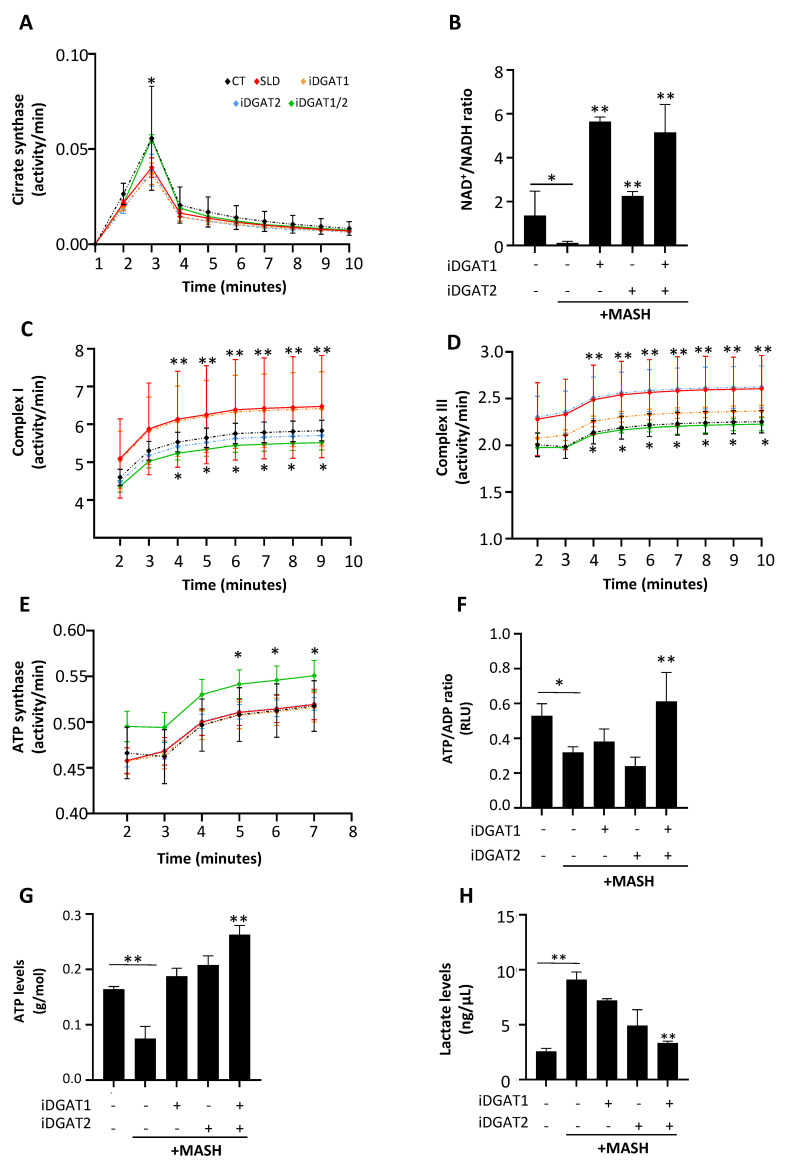
iDGAT1/2 recovered mitochondrial activity and improved ATP synthesis in HepG2 cells under MASH-like conditions. Evaluation of TCA was performed by assessing citrate synthase (CS) activity (**A**) over time and measuring NAD+/NADH ratios (**B**) in HepG2 cells challenged with MASH medium with or without DGAT inhibitors alone or in combination. Mitochondrial respiration rate of HepG2 cells was evaluated through the assessment of complex I (**C**), complex III (**D**) and ATP synthase (**E**) activities. ATP/ADP production was estimated in live HepG2 cells (**F**), while total ATP (**G**) and lactate (**H**) contents were quantified in HepG2 cell lysates after the exposure to MASH medium and/or DGAT inhibitors. Each experiment was carried out in triplicate. Adjusted * *p* < 0.05 or ** *p* < 0.01.

**Figure 5 ijms-25-09074-f005:**
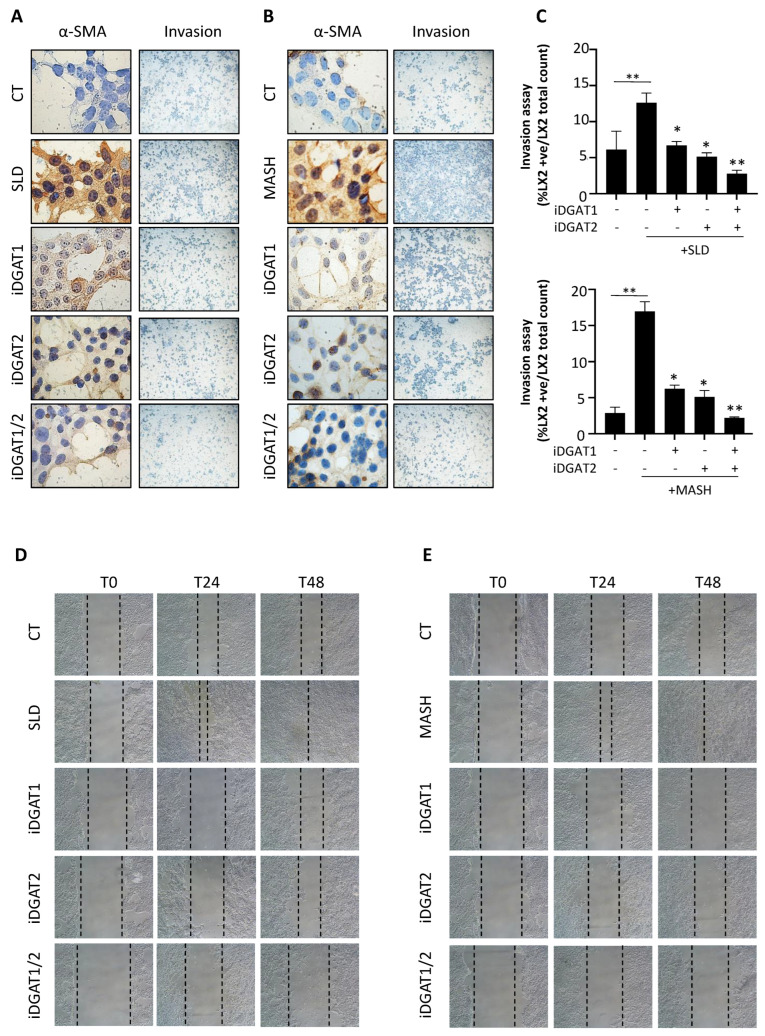
**iDGAT1 and/or iDGAT1 hinder LX2 cell activation after stimulation with SLD and MASH conditioned media.** (**A**,**B**) Representative images obtained from immunocytochemistry (40× magnification) of α-sma protein and from an invasion assay in LX2 cells receiving SLD and MASH conditioned media alone or supplemented with DGAT inhibitors. (**C**) Quantification of the invasion assay images was performed by counting LX2 positive nuclei (blue) in five random micrographs (40× magnification) and normalized to the total count of LX2 cells. (**D**,**E**) Wound-healing assay images were acquired at 0, 24, and 48 h (10× magnification). The dotted lines indicate the scratch width. At least three independent experiments were conducted. Adjusted * *p* < 0.05 or ** *p* < 0.01.

**Figure 6 ijms-25-09074-f006:**
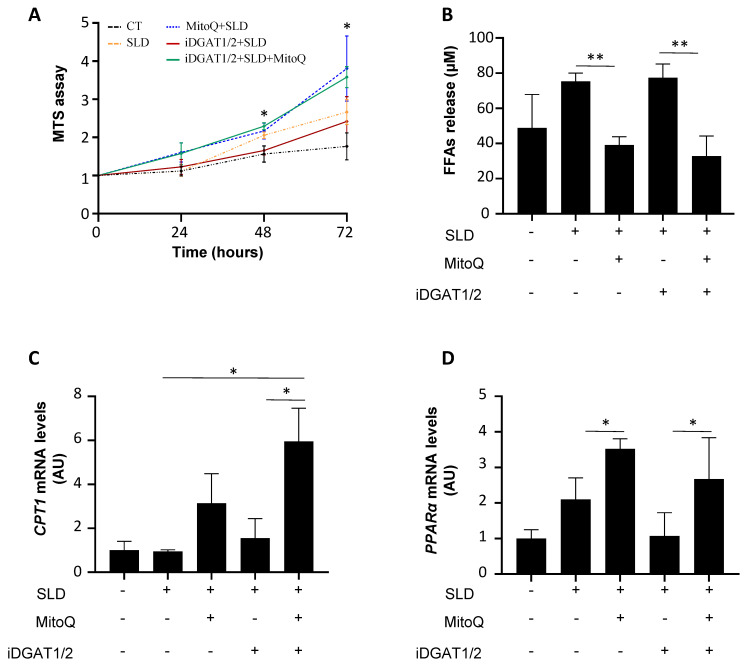
MitoQ enhanced the response to iDGAT1/2 by promoting FFA disposal in HepG2 cells under SLD medium. (**A**) Cell growth was assessed through MTS assay for 24, 48, and 72 h, and 1 week (λ = 490 nm). (**B**) FFAs were measured in cell supernatants after the exposure of HepG2 cells to SLD-medium with or without iDGAT1/2 at 25 µM and/or MitoQ at 1 µM. CPT1 (**C**) and PPARα mRNA levels (**D**) were assessed through qRT-PCR in HepG2 cells treated with SLD medium and/or DGAT inhibitors and/or MitoQ. Each experiment was carried out in triplicate. * *p* < 0.05 or ** *p* < 0.01.

**Table 1 ijms-25-09074-t001:** List of primers for qRT-PCR.

Gene	Primer forward 5′ → 3′	Primer Reverse 3′ → 5′
ATF4	AAACCTCATGGGTTCTCCAG	GGCATGGTTTCCAGGTCACT
Col1A1	CCATCAAAGTCTTCTGCAACATG	CGCCATACTCGAACTGGAATC
DGAT1	GCTTCAGCAACTACCGTGGCAT	CCTTCAGGAACAGAGAAACCACC
DGAT2	TCCAGCTGGTGAAGACACAC	GCTGACAGGGCAGATACCTC
MMP4	CCTTGGACTGTCAGGAATGAGG	TTCTCCGTGTCCATCCACTGGT
MMP9	GCCACTACTGTGCCTTTGAGTC	CCCTCAGAGAATCGCCAGTACT
MnSOD2	CAAATTGCTGCTTGTCCAAA	TCTTGCTGGGATCATTAGGG
SREBP1	TGCATTTTCTGACACGCTTC	CCAAGCTGTACAGGCTCTCC
SREBP2	CTCCATTGACTCTGAGCCAGGA	GAATCCGTGAGCGGTCTACCAT
TIMP1	TTTTGTGGCTCCCTGGAACA	AAACAGGGAAACACTGTGCAT
TIMP2	ACCCTCTGTGACTTCATCGTGC	GGAGATGTAGCACGGGATCATG
XBP1	GAAGCCAAGGGGAATGAAGT	GCCCAACAGGATATCAGACTC
β-actina	GCTACAGCTTCACCACCACA	AAGGAAGGCTGGAAAAGAGC

**Table 2 ijms-25-09074-t002:** List of antibodies used for WB and immunocytochemistry analyses.

Primary Antibody	Cat. Number *	Secondary Antibody	Cat. Number
DGAT1 (1:500)	Sigma-Aldrich SAB4301075	Anti-IgG Rabbit (1:5000)	Cell signaling #7076
DGAT2 (1:1000)	Sigma-ldrich SAB2106887	Anti-IgG Rabbit (1:5000)	Cell signaling #7076
α-SMA (1:150)	Cell Signaling #77397	Anti-IgG Rabbit (1:500)	Cell signaling #7076
Vinculin (1:1000)	Abcam ab73412	Anti-IgG Rabbit (1:5000)	Cell signaling #7076

* Sigma-Aldrich, St. Louis, MO, USA; Cell signaling, Danvers, MA 01923, USA; Abcam, Cambridge, UK.

## Data Availability

The original contributions presented in the study are included in the article/Appendix A, further inquiries can be directed to the corresponding author.

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
