# Peer review of "DGAT1 and DGAT2 Inhibitors for Metabolic Dysfunction-Associated Steatotic Liver Disease (MASLD) Management: Benefits for Their Single or Combined Application"

_ijms, 2024, doi:10.3390/ijms25169074_

Round 1
Reviewer 1 Report
Comments and Suggestions for Authors
First of all, Authors themselves said that it is in vitro model - most patients with metabolic disorders take different additional druga which may influence the efficacy of DAG
There is lack of proper histologic description of livers: the severity of steatosis (percentage); fibrosis (What scale if any was used to describe its advanced)
Microscopic pictures are not very clear to me
Comments on the Quality of English Language
Minor mistakes
Author Response
REVIEWER 1
First of all, Authors themselves said that it is in vitro model - most patients with metabolic disorders take different additional drug which may influence the efficacy of DGAT
We agree with the Reviewer's statement and, in response, have reviewed available data on pharmacological interactions between DGAT inhibitors and drugs used to manage metabolic disorders. Kulmatycki et al. previously reported that Pradigastat, a DGAT1 inhibitor, does not exhibit significant pharmacokinetic interactions with rosuvastatin, a cholesterol-lowering drug commonly used to treat dyslipidemia, a frequent comorbidity in MASLD patients (PMID: 25740267). The National Institutes of Health (NIH) also confirmed that Pradigastat has minimal drug-drug interactions, with evidence showing that while it may slightly reduce the excretion of rosuvastatin, the latter does not affect Pradigastat's efficacy (https://pubchem.ncbi.nlm.nih.gov/compound/Pradigastat). Similarly, Ervogastat, a DGAT2 inhibitor, has demonstrated a safe pharmacokinetic profile in clinical studies (https://pubchem.ncbi.nlm.nih.gov/compound/Ervogastat). Aarti Sawant-Basak and collaborators found that when Ervogastat was co-administered with Clesacostat, an acetyl-CoA carboxylase inhibitor, there were no clinically meaningful pharmacokinetic interactions (PMID: 38362827). These findings suggested that both DGAT inhibitors can be safely co-administered with other drugs commonly used to manage metabolic disorders without significant alterations in their pharmacological profiles.
There is lack of proper histologic description of livers: the severity of steatosis (percentage); fibrosis (What scale if any was used to describe its advanced)
We apologize for the misunderstanding. As the histological evaluation is not a fundamental part of our results, given that the study is entirely carried out in vitro, we did not deep into the details of the histological evaluation for all studies mentioned in the introduction and discussion.
In general, in preclinical models, steatosis was assessed through hematoxylin and eosin (H&E) staining, while fibrosis was evaluated based on collagen deposition using immunohistochemistry (IHC) or Sirius Red staining. For patients, most studies have assessed liver pathology using FibroScan and through biochemical monitoring of transaminase levels. We hope this clarification addresses your concern.
Microscopic pictures are not very clear to me
We attempted to improve the quality of the images.
Reviewer 2 Report
Comments and Suggestions for Authors
My main concern:
1. Does inhibiting iDGAT1and iDGAT2 affect normal liver cells?. The authors should provide MTT assay results for the effects on QSG-7701 normal liver cells cells.
2. What is the concentration of iDGAT1 and iDGAT2 repectively? It should be written in the article.
3.Figure 3A NASH should be changed to MASH
Author Response
REVIEWER 2
My main concern:
- Does inhibiting iDGAT1and iDGAT2 affect normal liver cells? The authors should provide MTT assay results for the effects on QSG-7701 normal liver cells.
We thank the reviewer for the comment. Previously, it has been shown that inhibition of DGAT1 and DGAT2 in primary hepatocytes exerts complementary roles in lipid metabolism (PMID: 25792450). DGAT1 supports fatty acid oxidation and lipid droplet expansion, while DGAT2 is crucial for VLDL secretion and de novo lipogenesis. Moreover, DGAT1 and DGAT2 can compensate for each other in synthesizing triglycerides. Indeed, when one enzyme is inhibited or inactivated, the other can partially take over its function to ensure continued TG synthesis and maintain lipid homeostasis, thus suggesting that inhibiting DGAT1 and DGAT2 may not adversely affect normal liver cells.
We appreciate the suggestion to provide MTT in QSG-7701 normal liver cells. Unfortunately, the QSG-7701 are not available in our laboratory, and we will not be able to obtain them within the 5-day revision period.
- What is the concentration of iDGAT1 and iDGAT2 repectively? It should be written in the article.
We apologize for the inaccuracy. The iDGAT1 and iDGAT2 concentrations have now been reported in the materials and methods sections (in yellow)
3.Figure 3A NASH should be changed to MASH
We apologize for the inaccuracy. We replaced NASH with MASH in Figure 3A
Round 2
Reviewer 2 Report
Comments and Suggestions for Authors
The author has been improvements to fault.